# Compositionally aware estimation of cross-correlations for microbiome data

Ib Thorsgaard Jensen[1,2]*, Luc Janss[3], Simona Radutoiu[1], Rasmus Waagepetersen[2]*

**1** Department of Molecular Biology and Genetics, Aarhus University, Aarhus, Denmark, **2** Department of Mathematical Sciences, Aalborg University, Aalborg, Denmark, **3** Center for Quantitative Genetics and Genomics, Aarhus University, Aarhus, Denmark

* itj@math.aau.dk (ITJ); rw@math.aau.dk (RW)

**Data Availability Statement:** All data used in this paper can be found at https://github.com/IbTJensen/Microbiome-Cross-correlations/. The raw sequencing data from Byrd et al. can be found in NCBI Bioproject 46333, and the OTU table was

## Abstract

In the field of microbiome studies, it is of interest to infer correlations between abundances of different microbes (here referred to as operational taxonomic units, OTUs). Several methods taking the compositional nature of the sequencing data into account exist. However, these methods cannot infer correlations between OTU abundances and other variables. In this paper we introduce the novel methods SparCEV (Sparse Correlations with External Variables) and SparXCC (Sparse Cross-Correlations between Compositional data) for quantifying correlations between OTU abundances and either continuous phenotypic variables or components of other compositional datasets, such as transcriptomic data. SparCEV and SparXCC both assume that the average correlation in the dataset is zero. Iterative versions of SparCEV and SparXCC are proposed to alleviate bias resulting from deviations from this assumption. We compare these new methods to empirical Pearson cross-correlations after applying naive transformations of the data (log and log-TSS). Additionally, we test the centered log ratio transformation (CLR) and the variance stabilising transformation (VST). We find that CLR and VST outperform naive transformations, except when the correlation matrix is dense. SparCEV and SparXCC outperform CLR and VST when the number of OTUs is small and perform similarly to CLR and VST for large numbers of OTUs. Adding the iterative procedure increases accuracy for SparCEV and SparXCC for all cases, except when the average correlation in the dataset is close to zero or the correlation matrix is dense. These results are consistent with our theoretical considerations.

## Introduction

Sequencing data are ubiquitous in modern biology [1]. For example, RNA-seq data have been used to identify genes associated with clinical outcomes of cancer patients [2], for human disease profiling [3], and to identify genes with possible links to Rett Syndrome [4]. Microbiome data have drawn much attention in recent years, particularly regarding the human gut microbiome. Composition of the human gut microbiome has been shown to be associated with several aspects of human health, such as obesity [5] and metabolic disorders [6]. More recently,

originally obtained from Morton et al. at https://github.com/knightlab-analyses/reference-frames. The raw sequencing data from Thiergart et al. can be found at the European Nucleotide Archive (ENA). The 16S dataset has project accession no. PRJEB34100, and the ITS dataset has project accession no. PRJEB34099. The OTU tables was originally obtained at https://github.com/ththi/Lotus-Symbiosis.

**Funding:** This work was supported by the Bill and Melinda Gates Foundation and from Foreign, Commonwealth & Development Office through Engineering the Nitrogen Symbiosis for Africa (ENSA; OPP11772165). Ib Thorsgaard Jensen and Rasmus Waagepetersen were supported by research grant VIL57389 from Villum Fonden. The funders played no role in the content of this paper.

**Competing interests:** The authors have declared that no competing interests exist.

the integration of microbiome data with other omics data has received increasing interest [7–11].

Data from sequencing technologies pose many specific challenges. They produce count data with technical noise, which makes results from rare features difficult to interpret. Additionally, they are compositional, meaning that the observed variables are components of an arbitrary total. As a result, apparent correlations involving sequencing data, such as microbiome data, may be due to the technical constraints and not biologically meaningful. We refer to this phenomenon as *compositional effects*. For datasets with few variables, direct examination of correlations might seem appealing. However, compositional effects become more pronounced in case of few variables and hence direct examination of the observed data does not suffice. Instead, it is pertinent to develop statistical methods that correct for compositional effects.

Within the field of microbiome studies, several methods have been proposed to infer interactions between microbial abundances. These include Local Similarity Analysis [12], which finds non-linear relationships through time using time series data; Sparse Compositional Correlations (SparCC), which infers correlations based on compositional data [13]; and Sparse Inverse Covariance Estimation for Ecological Association Inference (SPIEC-EASI), which infers relations through graphical models [14]. In this paper, we focus on the estimation of correlation coefficients.

When considering correlations in the context of compositional data, there are essentially three cases of interest: A) correlations between features of the same compositional dataset, B) cross-correlations between features of a compositional dataset and non-compositional variables, and C) cross-correlations between two compositional datasets. Correlations between bacterial abundances in a microbiome is an example of case A. An example of case B is cross-correlations between gut microbes and clinical features of patients [15], and an example of case C is cross-correlations between microbial abundances and gene expression levels from RNA-seq data [7]. For an overview of these cases, see Table 1.

The methods mentioned above all operate in case A. Case B appears not have received much attention so far while more recently, case C has gained more interest, with new methods being developed. For example, SPIEC-EASI has been extended to infer interactions between variables from two compositional datasets [16]. Like the original SPIEC-EASI, pairs of variables that are conditionally independent are identified by estimating the precision matrix using a penalized estimation scheme to enforce sparsity. The method mmVec [17] is designed for identifying interactions between OTU abundances and metabolite concentrations. This

**Table 1. Cases A, B, and C along with explanations and and overview of applicable methods.**

| Case | Correlations | Methods |
|---|---|---|
| A | Within a composition | SparCC |
| | | SPIEC-EASI |
| | | Local Similarity Analysis |
| | | Pearson |
| B | Between a composition and an external variable | SparCEV |
| | | Pearson |
| C | Between two compositions | SparXCC |
| | | SPIEC-EASI |
| | | mmVec |
| | | Pearson |

method employs a neural network architecture to estimate the probability of observing a metabolite, given that a specific OTU is observed. It was shown to perform with similar accuracy as the extended SPIEC-EASI and to outperform correlation-based procedures. However, the correlations were estimated using a flawed methodology, where the centered log-ratio (CLR) transformation was applied to both datasets simultaneously, rather than separately, thus biasing the results. Quinn and Erb [18] showed that when the CLR-transformation was applied appropriately, correlation-based methods outperform both mmVec and SPIEC-EASI in the setup examined by Morton et al. [17]. In response, Morton et al. showed that it was possible to construct scenarios where mmVec outcompeted all alternatives [19]. In conclusion, there is scope for further developing and investigating methods for cases B and C.

In this paper, we focus on inferring cross-correlations in cases B and C. We introduce two novel compositionally aware methods, SparCEV (Sparse Correlations with External Variables) and SparXCC (Sparse Cross-Correlations between Compositional data). Using simulation studies, we compare these methods to Pearson cross-correlations applied to various transformations of the data. Theoretical comparisons of transformation-based methods and derivations of new methods are given in the supplementary material.

## Materials and methods

### Modelling sequencing data

Let $a_i$ denote the absolute abundance of OTU $i$, $i = 1, \ldots, p$, $A = \sum_{j=1}^{p} a_j$, and $r_i = a_i/A$ the relative abundance. The aim of this paper is to estimate the correlation between $\log a_i$ and other log transformed variables. However, we only have access to observed read counts, denoted $x_i$ for OTU $i$. To theoretically compare the different strategies and to develop new methods, we adopt a simplified modelling framework, where

$$x_i = r_i N, \tag{1}$$

where $N = \sum_{j=1}^{p} x_j$ denotes the library size. We compare the methods considered using simulations from models that are more complex and realistic than (1), see the section "Simulation models". In these models, $x_i$ given $(r_i, N)$ is not fixed. We use the term *technical variance* for the variance of $x_i$ given $(r_i, N)$ and the term *biological variance* for the variance of $r_i$. The more realistic models are, however, intractable for theoretical analysis.

All tested methods but one require log-transformation of the $x_i$s, which is problematic if $x_i = 0$ is observed. As a remedy, we add 1 to all read counts prior to log-transformation of the data (the pseudo-count method).

### Existing strategies for cross-correlation estimation

**Naive transformation.** We use the term *naive transformation* to refer to any transformation that does not take the compositional nature of the data into account. Naive transformations considered in this paper are log and log total sum scaling (TSS). Theoretically, naive transformations do not adequately account for the compositional structure of the data (see S1 Text). Nonetheless, it remains a common practice to apply these transformations [20] or no transformations at all [10, 15, 21–24]. As a result, any method that outperform naive transformations would constitute an improvement relative to common practice.

**Adapted transformations.** We use the term *adapted transformations* to refer to transformations that are adapted to the particular structure of the data beyond differing library sizes. In this paper, we consider the centered log-ratio (CLR) and the variance-stabilising transformation (VST). See Table 2 for definitions of all transformations (naive and adapted)

**Table 2. An overview of the transformations used to assess cross-correlations.** In VST, $k$ is a replicate index.

| Transformation | Expression | Interpretation |
|---|---|---|
| log | $\log x_i$ | Log-transformed observed read counts |
| log-TSS | $\log \mathrm{TSS}(x_i) = \log\left(\frac{x_i}{\sum_{j=1}^{p} x_j}\right)$ | The log of estimated relative abundances |
| CLR | $\mathrm{CLR}(x_i) = \log x_i - \frac{1}{p}\sum_{j=1}^{p}\log x_j$ | Abundances relative to the average abundance |
| VST | $\mathrm{VST}(x_{ik}) = \int_0^{x_{ik}/\hat{s}_k} f(\mu)\mathrm{d}\mu, \text{ where } f(\mu) = \mu^2\hat{a}_0 + \mu(1 + \hat{a}_1)$ | Removal of mean-variance relationship |

considered in this paper. Some common transformations, such as trimmed M-means [25], DESeq's median-based transformation [26], and upper-quartile transformation [27], are not included, since they do not correct for within-replicate biases and are thus not applicable for correlation estimation.

VST makes use of the DESeq modelling framework [26]. Specifically, it assumes that $x_{ik} \sim$ NB$(\mu_{ik}, \phi_i)$, where $\mu_{ik} = s_k\lambda_i$ and $\phi_i = a_0 + a_1/\lambda_i$. The estimates $\hat{a}_1$, $\hat{a}_0$, $\hat{\lambda}_i$ and $\hat{s}_k$ are obtained using the DESeq2 estimation procedures [26, 28]. Here, $k$ denotes the index of the biological replicate. VST requires at least one feature without any zero counts. This may be violated for data with small $p$ which is frequently the case for our simulated microbiome data. We therefore only consider the VST transformation in case C for simulated gene expression data while applying CLR to the microbiome data.

For convenience, we use the term CLR for the method where empirical Pearson cross-correlations are applied to CLR-transformed data, and likewise for log, log-TSS, and VST.

**Theoretical assessment of transformations.** We examine theoretically whether empirical Pearson cross-correlations combined with the transformations presented in Table 2 are likely to yield good approximations of cross-correlations. We focus on case B, since it is simpler and the results in case C are analogous. We seek an approximation of $\mathbb{C}\text{orr}\left[\log a_i, b\right]$, where $b$ is a non-compositional variable, here referred to as a *phenotypic variable*. By (1) and Table 2 we have,

$$x_i = \frac{a_i}{A}N, \quad \mathrm{TSS}(x_i) = \frac{a_i}{A}, \quad \mathrm{CLR}(x_i) = \log a_i - \frac{1}{p}\sum_{j=1}^{p}\log a_j.$$

By definition,

$$\mathbb{C}\text{orr}\left[\log a_i, b\right] = \frac{\mathbb{C}\text{ov}\left[\log a_i, b\right]}{\sqrt{\mathbb{V}\text{ar}\left[\log a_i\right]}\sqrt{\mathbb{V}\text{ar}\left[b\right]}}, \tag{2}$$

and we require good approximations of the numerator and the denominator. Since $b$ is not compositional, $\mathbb{V}\text{ar}\left[b\right]$ can easily be estimated without the need for approximation. According to our derivations in S1 Text, log, log-TSS, and CLR all lead to reasonable approximations of the covariance $\mathbb{C}\text{ov}\left[\log a_i, b\right]$ under the model in (1) given appropriate assumptions. Furthermore, we show that

$$\mathbb{V}\text{ar}\left[\mathrm{CLR}\left(x_i\right)\right] \approx \mathbb{V}\text{ar}\left[\log a_i\right]$$

when $p$ is large. However, analogous results do not hold for log and log-TSS, demonstrating that naive transformations are not sufficient. Summing up, CLR can yield good approximations under the model in (1) with the following assumptions:

(Bi) $\frac{1}{p}\sum_{j\neq i}\mathbb{C}\text{ov}\left[\log a_i, \log a_j\right] \approx 0$ for all $i$

(Bii)$\frac{1}{p} \sum_{i=1}^{p} \mathbb{C}ov\left[\log a_i, b\right] \approx 0$ for all $i$

(Biii) The number of OTUs, $p$, is large.

In case C, we we need (Bi), (Biii), and the additional assumptions

(Ci)$\frac{1}{q} \sum_{l \neq k} \mathbb{C}ov\left[\log b_k, \log b_l\right] \approx 0$ for all $k$

(Cii)$\frac{1}{q} \sum_{l=1}^{q} \mathbb{C}ov\left[\log a_i, b_l\right] \approx 0$ for all $i$

(Ciii)$\frac{1}{p} \sum_{j=1}^{p} \mathbb{C}ov\left[\log a_j, b_k\right] \approx 0$ for all $k$

(Civ) The number of genes, $q$, is large

Conditions (Bi), (Bii) and (Ci)-(Ciii) hold if the correlation matrix is sparse; thus, following the language of Friedman and Alm [13], we refer to these as *sparsity assumptions*. This is a slight abuse of terminology, since these conditions may also hold if all rows of the correlation matrix contain entries whose distributions are symmetric around zero, even though such a matrix is not sparse.

## Compositionally aware methods

Inspired by SparCC, we introduce compositionally aware methods for cases B and C. In case B, we assume the same sparsity condition as for CLR in the previous section. We then show in S1 Text that

$$\mathbb{C}orr\left[\log a_i, b\right] \approx \frac{1}{\sigma_b \alpha_i} \frac{1}{p-1} \sum_{j \neq i} \mathbb{C}ov\left[\log \frac{x_i}{x_j}, b\right],$$

where $\alpha_i^2 = \mathbb{V}ar\left[\log a_i\right]$ can be estimated by SparCC and $\sigma_b^2 = \mathbb{V}ar\left[b\right]$ can be estimated in a standard fashion. In contrast to CLR this method only requires conditions (Bi) and (Bii), but not (Biii). Therefore, it is likely preferable when $p$ is small. We name this method ***SparseCorrelations of External Variables*** (SparCEV).

In Case C, we let $b_k$, $k = 1, \ldots, q$, denote the gene expression level of the $k$th gene, $B = \sum_{k=1}^{q} b_k$, and $M$ the library size. Similar to (1), we assume the model

$$y_k = \frac{b_k}{B} M$$

for the observed gene expression level. In S1 Text, we obtain the relation

$$\mathbb{C}orr\left[\log a_i, \log b_k\right] \approx \frac{t_{ik}}{(p-1)(q-1)\alpha_i \beta_k}, \tag{3}$$

where

$$t_{ik} = \sum_{j \neq i} \sum_{l \neq k} \mathbb{C}ov\left[\log \frac{x_i}{x_j}, \log \frac{y_k}{y_l}\right].$$

The parameters $\alpha_i^2 = \mathbb{V}ar\left[\log a_i\right]$ and $\beta_k^2 = \mathbb{V}ar\left[\log b_k\right]$ can be approximated by applying SparCC to the microbiome and gene expression datasets individually, and the variances in $t_{ik}$ can be estimated in a standard fashion. Details regarding efficient computation of the $t_{ik}$s are given in the S1 Text. As with CLR, we need the assumptions (Bi) and (Ci)-(Ciii), but unlike

CLR, we do not need (Biii) and (Civ). We refer to this method as **Sparse Cross-Correlations of Compositional data** (SparXCC, where "X" represents "cross").

**Iterative procedures.** Unless all OTUs are uncorrelated with the non-compositional variables (case B) or the compositional variables of the other dataset (case C), the estimates above are biased. Specifically, in case B, we get

$$\frac{1}{p-1} \sum_{j \neq i} \mathbb{C}\mathrm{ov}\left[\log \frac{x_i}{x_j}, b\right] = \mathbb{C}\mathrm{ov}\left[\log a_i, b\right] - \frac{1}{p-1} \sum_{j \neq i} \mathbb{C}\mathrm{ov}\left[\log a_j, b\right].$$

Estimates based on the right-hand side are useful under the assumption that the second term on the right-hand side is small. However, in practice this assumption may be violated leading to estimation bias.

Now suppose we could identify the set $R = \{j : \mathbb{C}\mathrm{orr}\left[\log a_j, b\right] = 0\}$. Then,

$$\frac{1}{|R|} \sum_{j \in R} \mathbb{C}\mathrm{ov}\left[\log \frac{x_i}{x_j}, b\right] = \mathbb{C}\mathrm{ov}\left[\log a_i, b\right].$$

This observation motivates an iterative procedure, similar to the one employed by SparCC in case A. For iteration $n$, we estimate

$$\hat{\rho}_i^{(n)} = \frac{1}{\hat{\sigma}_b \hat{\alpha}_i} \frac{1}{|R_n|} \sum_{j \in R_n} \hat{C}_{ij},$$

where $\hat{C}_{ij}$ and $\hat{\sigma}_b$ are the standard empirical estimates of $\mathbb{C}\mathrm{ov}\left[\log \frac{x_i}{x_j}, b\right]$ and $\mathbb{V}\mathrm{ar}\left[b\right]$ respectively, $\hat{\alpha}_i$ is the SparCC estimate of $\mathbb{V}\mathrm{ar}\left[\log a_i\right]$, and

$$R_n = \{i : |\hat{\rho}_i^{(n-1)}| < t\}, \tag{4}$$

where $t$ is some user-specified threshold. To initialize, we set $R_0 = \{1, \ldots, p\}$, and iteration concludes once $R_n = R_{n-1}$. When a distinction is necessary, we refer to SparCEV with and without the iterative procedure as SparCEV base and SparCEV iterative respectively. Unless otherwise stated, SparCEV refers to SparCEV iterative.

In case C, we have

$$\frac{t_{ik}}{(p-1)(q-1)} = \mathbb{C}\mathrm{ov}\left[\log a_i, \log b_k\right] - \frac{1}{p-1} \sum_{i \neq j} \mathbb{C}\mathrm{ov}\left[\log a_j, \log b_k\right]$$
$$- \frac{1}{q-1} \sum_{l \neq k} \mathbb{C}\mathrm{ov}\left[\log a_i, \log b_l\right] + \sum_{j \neq i} \sum_{l \neq k} \mathbb{C}\mathrm{ov}\left[\log a_j, \log b_l\right].$$

In order to eliminate the last three terms, we need the sets $S = \{i : \rho_{ik} = 0 \text{ for all } k\}$ and $T = \{k : \rho_{ik} = 0 \text{ for all } i\}$. Then,

$$\frac{1}{(p-1)(q-1)\alpha_i \beta_k} \sum_{j \in S} \sum_{l \in T} \mathbb{C}\mathrm{ov}\left[\log \frac{x_i}{x_j}, \log \frac{y_k}{y_l}\right] = \mathbb{C}\mathrm{orr}\left[\log a_i, \log b_k\right].$$

In iteration $n$, we estimate

$$\hat{\rho}_{ik}^{(n)} = \frac{1}{(p-1)(q-1)\hat{\alpha}_i \hat{\beta}_k} \sum_{j \in S_n} \sum_{l \in T_n} \hat{C}_{ijkl},$$

where $\hat{C}_{ijkl}$ is the standard empirical estimate of $\mathbb{C}\mathrm{ov}\left[\log \frac{x_i}{x_j}, \log \frac{y_k}{y_l}\right]$, $\hat{\alpha}_i$ and $\hat{\beta}_k$ are the SparCC

estimates of $\mathbb{V}\mathrm{ar}\left[\log a_i\right]$ and $\mathbb{V}\mathrm{ar}\left[\log b_k\right]$ respectively, and

$$S_n = \left\{ i : \frac{1}{q} \sum_{l=1}^{q} |\hat{\rho}_{il}^{(n-1)}| < t_1 \right\}, \quad T_n = \left\{ k : \frac{1}{p} \sum_{j=1}^{p} |\hat{\rho}_{jk}^{(n-1)}| < t_2 \right\},$$

where $t_1$ and $t_2$ are user-specified thresholds. Similar to case B, we set $S_0 = \{1, \ldots, p\}$ and $T_0 = \{1, \ldots, q\}$, and iterate until $T_n = T_{n-1}$ and $S_n = S_{n-1}$. We use the terminology SparXCC base and SparXCC iterative analogously to SparCEV base and SparCEV iterative.

For some datasets, there may be no bias to correct and in such cases SparXCC base or SparCEV base are appropriate. To assess this in practice, one may estimate the correlation coefficients with both the base and iterative versions and then plot them against each other. The bias terms for each correlation coefficient are approximately identical, at least when $p$ (and $q$ in case C) are large. Thus, the discrepancies between the base estimates and the iterative estimates should be similar for all $i$ (and $k$ in case C). Consequently, if the pairs of estimates are far from a straight line with slope 1, this indicates that the iterative procedure does not produce useful estimates. This may happen, for example, when the threshold is too low, whereby too few OTUs (and/or genes) are included. On the other hand, if the pairs of estimates are close to a straight line with slope 1 and intercept different from zero, this indicates that the iterative procedure succeeds in correcting for the bias and should thus be used.

**Choice of threshold.** The thresholds $t$, $t_1$, and $t_2$ are important when carrying out the iterative versions of SparCEV and SparXCC. If they are set too low, few variables may qualify and the estimates in the subsequent iterations may become unreliable. If they are set too high, we risk including highly correlated OTUs (and genes in case C) which also renders the estimates less reliable. To address this, we employ a bootstrap procedure to select the thresholds. In case B, we permute the non-compositional variable, thus breaking the correlation with all OTUs. Then, we use SparCEV base on this permuted dataset to obtain the set of estimates

$$M_{\mathrm{Perm}} = \{|\hat{\rho}_1^{\mathrm{Perm}}|, \ldots, |\hat{\rho}_p^{\mathrm{Perm}}|\}.$$

We can ensure that the vast majority of uncorrelated OTUs are in $R_1$ (cf. (4)) by choosing $t = \max\{M_{\mathrm{Perm}}\}$, but we may still risk including correlated pairs. Alternatively one may use some percentile of $M^{\mathrm{Perm}}$. We use the 80th percentile by default, but the results should always be examined relative to the base version as described above. If the results are far from a straight line with slope 1, the user may wish to experiment with different choices of $t$, such as different percentiles of the bootstrap set.

## Simulation models

We adopt the parametric model employed by *SparseDOSSA2* [29], adapting the methodology slightly to handle cases B and C. A simulated dataset contains $n \geq 1$ replicates, where $n$ is the number of microbiome samples sequenced. The individual simulated variables (e.g. abundances or gene expression levels) are characterized by the mean, $\mu_i$, variance, $\sigma_i^2$, and zero-probability, $\pi_i$. The parameter $\pi_i$ reflects the probability that OTU $i$ is absent from a given replicate. We refer to this as a *biological zero*. The correlation between variables is characterized by the correlation matrix $\Psi$. We simulate $p$ OTU abundances, $a_1, \ldots, a_p$ and $q$ other variables, $b_1, \ldots, b_q$. In case B, the latter $q$ variables are non-compositional, typically $q = 1$, and we take $\pi_{p+k} = 0$, $k = 1, \ldots, q$, so that biological zeros do not occur for the $b_k$s. In case C, $q > 1$ and the $b_k$s are compositional. The library size $N^a$ is simulated from a log normal distribution with parameters $\mu_a$ and $\sigma_a^2$.

The simulation algorithm is given in the following steps. For ease of presentation, we present the case where $n = 1$, but when $n \geq 1$ the steps would simply be repeated $n$ times.

1. Simulate the $p + q$-dimensional variable $g \sim \mathrm{N}(0, \Psi)$.

2. Define the variables $Z_i$ for $i = 1, \ldots, p + q$ such that $Z_i = 0$ if $g_i < \Phi^{-1}(\pi_i)$ and $Z_i = F_i^{-1}(\Phi(g_i))$ otherwise, where $\Phi$ is the standard normal cumulative distribution function (cdf) and $F_i(t) = \pi_i + (1 - \pi_i)\Phi((\log t - \mu_i)/\sigma_i^2)$ is the cdf of a zero-inflated log-Gaussian distribution with parameters $(\pi_i, \mu_i, \sigma_i^2)$. We now have

$$\log Z_i | Z_i \neq 0 \sim \mathrm{N}(\mu_i, \sigma_i^2) \text{ and } P(Z_i = 0) = \pi_i.$$

3. Set $a_i = Z_i$ for $i = 1, \ldots, p$ as the absolute OTU abundances. In case C, $b_j = Z_{j+p}$ for the absolute gene expression levels. In case B, we let $b_j = \log Z_{j+p}$ for the non-compositional phenotypic variables.

4. Set $r_i^a = a_i / \sum_{k=1}^p a_k$ as the relative abundances of the OTUs, and in case C, set $r_j^b = b_j / \sum_{k=1}^q b_k$ as the relative expression levels.

5. Simulate $N^a \sim \log \mathrm{N}(\mu_a, \sigma_a^2)$ and let $\lceil N^a \rceil$ be the library size.

6. Simulate the vector, $x = (x_1, \ldots, x_p)^\top$, of observed read counts of OTU $1, \ldots, p$ as $x \sim \mathrm{Multinom}\left(\lceil N^a \rceil, r_1^a, \ldots, r_p^a\right)$.

7. In case C, simulate $N^b \sim \log \mathrm{N}(\mu_b, \sigma_b^2)$ and let $\lceil N^b \rceil$ be the library size.

8. In case C, simulate the vector, $y = (y_1, \ldots, y_q)^\top$, of observed read counts of gene $j = 1, \ldots, q$ as $y \sim \mathrm{Multinom}\left(\lceil N^b \rceil, r_1^b, \ldots, r_p^b\right)$.

In case B, steps 7–8 are skipped. Excluding the simulation of the $b_j$s, steps 1–6 in the above procedure are identical to the procedure employed by SparseDOSSA2 [29]. The above simulation scheme differs from the model (1) by the multinomial noise generated in steps 6 and 8 where the multinomial model is a simplistic representation of the randomness generated in the sequencing procedure. The correlation $\mathbb{C}\mathrm{orr}\,[\log a_i, \log a_j]$ agrees with $\Psi_{ij}$ when $\pi_i = \pi_j = 0$. This does not hold in the presence of biological zeros, $\pi_i > 0$ or $\pi_j > 0$, in which case $\log a_i$ or $\log a_j$ may not even be well defined. We nevertheless use $\Psi$ as a ground truth for comparison with our estimates, including the case of biological zeros. In that way, the presence of biological zeros is considered a source of noise relative to our method in addition to the multinomial noise.

The correlation matrix $\Psi$ is constructed using two methods described in S1 Text. The first is called the cluster method, and it works by assigning a portion of the OTUs to a "cluster". All OTUs in the cluster are correlated to each other with the same correlation coefficient and uncorrelated to every other OTU. All OTUs outside the cluster are also uncorrelated with each other. In case C, a similar portion of the genes are also assigned to the cluster, and in case B, all non-compositional variables are also assigned to the cluster. This gives us a high degree of control over the degree of sparsity and the strength of the correlations. The second method is called the loadings method, and it results in a correlation matrix without exact zero entries but where most variables are only weakly correlated, with a relatively small proportion of highly correlated variable pairs. The loadings method most likely results in more realistic systems than the cluster method, but it suffers from the limitation that it tends to produce matrices whose entries are symmetric around zero. This need not be true in a natural system.

Throughout the simulations in this paper, we simulate $n = 50$ replicates. In many practical settings, $n$ is considerably lower than that. However, for the purposes of the present simulation study, it is important that we can detect biases in the estimators. If the bias is small relative to the variance of the estimator, it may be difficult to detect in a simulation study. Since the variance of an estimator increases as $n$ decreases, it is counter-productive to perform simulation studies with small $n$. In other words, we construct a situation where the main bottleneck to producing accurate results is the chosen method, not the size of the dataset.

**Selecting parameter values.** In the simulation studies carried out in this paper, the log-scale parameters, $\mu_i$ and $\sigma_i^2$, and zero-probabilities, $\pi_i$, are chosen using a real dataset as a template. We estimate the mean, $\mu_{ri}$, and variance, $\sigma_{ri}^2$ of the observed read counts for $i = 1, \ldots, p$. We then choose $\mu_i$ and $\sigma_i^2$ such that the simulated variables have mean $\mu_{ri}$ and variance $\sigma_{ri}^2$, on the linear scale. By the properties of the log-normal distribution, the means and variances are related by

$$\sigma_i^2 = \log\left(1 + \frac{\sigma_{ri}^2}{\mu_{ri}^2}\right), \quad \mu_i = \log \mu_{ri} - \frac{\sigma_i^2}{2}. \tag{5}$$

The parameters $\pi_i$ are set to half the proportion of zeros for the $i$th variable, with the assumption that half of the zeros are biological and the other half are technical. In case B, the parameters of the phenotype variable are somewhat arbitrarily chosen so that it has a mean of 30 and a variance of 1. For the microbiome data, we use a dataset by Tao et al. [30] as a template, and for the gene expression data we use a currently unpublished dataset. All parameters used in the simulations are available at https://github.com/IbTJensen/Microbiome-Cross-correlations/.

We also examine the impact of diversity on the accuracy of the correlation estimation methods. We measure diversity using the *effective number of OTUs*, $p_{\text{eff}}$. We have $p_{\text{eff}} = e^H$, where $H = \sum_{i=1}^{p} r_i \log r_i$ is the entropy or *Shannon index*. The quantity $p_{\text{eff}}$ can be interpreted as the minimal number of OTUs such that a replicate has entropy $H$. This occurs when all OTUs are equally abundant. Here, we choose $\sigma_i^2 = 1$ and $\pi_i = 0$ for $i = 1, \ldots, p$ and $\mu_i$ in such a way that we get a specific value of $p_{\text{eff}}$ in expectation. This is accomplished by selecting the linear-scale mean relative abundances $v_i = \frac{1 - v_1}{p - 1}$ for $i \geq 2$ and obtaining $v_1$ by solving the equation

$$\log p_{\text{eff}} = v_1 \log v_1 + \sum_{i=2}^{p} \frac{1 - v_1}{p - 1} \log \frac{1 - v_1}{p - 1} \tag{6}$$

for $v_1$ given a choice of $p_{\text{eff}}$. We then choose an arbitrary value for the microbial load, say 1000, and set $\mu_{ri} = 1000 v_i$. Finally, the $\mu_i$ are obtained from the right part of (5).

## Methods assessment

We might assess the accuracy of correlation estimates $\hat{\rho}_{ij}$ by comparing the true correlations to the estimated correlations by computing, for example, the mean absolute error (MAE). However, suppose we use the estimate $\hat{\rho}_{ij} = 0$ for all $i, j$ and pick, for example, $c = 0.05$ and $\rho = 0.75$ for the cluster method. Then, even though non-zero correlations are not well estimated, the MAE, $c\rho = 0.0375$, is quite low. Thus, we separately consider the MAE of the pairs whose true correlation is zero and the MAE of the pairs whose true correlation is non-zero. In case of the loadings method, no correlations are exactly zero, so we then instead assess the MAE of pairs whose true correlation-coefficient exceeds the thresholds 0, 0.1, ..., 0.8.

In summary, for the cluster method, we use the criteria

$$\frac{1}{|S|} \sum_{(i,j)\in S} |\Psi_{ij} - \hat{\rho}_{ij}|, \quad \frac{1}{pq - |S|} \sum_{(i,j)\notin S} |\Psi_{ij} - \hat{\rho}_{ij}|,$$

where $S = \{(i,j) \in A_{pq} | \Psi_{ij} \neq 0\}$ and $A_{pq} = \{(i,j) \in \mathbb{N}^2 | 0 \leq i \leq p, \ p < j \leq p + q\}$. For the loadings method, we use the criteria

$$\frac{1}{|S_u|} \sum_{(i,j)\in S_u} |\Psi_{ij} - \hat{\rho}_{ij}|, \ \text{ where } \ S_u = \{(i,j) \in A_{pq} : |\Psi_{ij}| \geq u\}, \ \text{ for } \ u = 0, 0.1, \ldots, 0.8, \quad (7)$$

where $u = 0$ corresponds to the overall MAE.

## Discriminating between correlated and uncorrelated pairs

Since sparsity is only an approximate assumption, any test-statistic used to derive $p$-values is likely to be biased. This is exacerbated by the technical noise, which has particularly high impact for low-abundance OTUs. We shall not attempt to remedy these challenges here. Instead of using $p$-values, we choose a dynamic threshold based on the data. Pairs whose estimated absolute correlation exceeds this threshold are considered the most likely candidates for genuinely correlated pairs. The threshold is derived in the following way. Let $X$ and $Y$ be the two datasets under study (in case C, $Y$ is compositional, in case B it is not). Permute each dataset separately. This breaks all cross-correlation, but not the correlations within each dataset. Let $S_{\text{perm}}$ be the set of cross-correlation estimates obtained from the permuted data and let

$$m = \max \{|\hat{\rho}_1^{\text{Perm}}|, \ldots, |\hat{\rho}_p^{\text{Perm}}|\},$$

where $\hat{\rho}_i^{\text{Perm}}$ is the correlation with OTU $i$ estimated by applying SparCEV (or SparXCC in case C) on the permuted data (in case C, replace indices $i$ with $ik$ where appropriate). We consider OTU $i$ to be correlated with the variable of interest if $|\hat{\rho}_i| > m$.

## Implementation and code availability

Illustrations are produced using ggplot2 version 3.4.1 [31], ggpubr version 0.6.0 [32], and GGally version 2.1.2 [33]. The VST transformation was performed using DESeq2 version 1.34.0 [28]. Correcting for differences between experimental groups were carried out with limma version 3.58.1 [34]. Running time was measured using microbenchmark version 1.4.10 [35]. Hypothesis testing on cross-correlations were carried out using psych version 2.2.9 [36]. The SPIEC-EASI networks were estimated using the package SpiecEasi version 1.1.2 [37]. Implementation of SparCEV and SparXCC (both base and iterative) are available in the R-package `CompoCor`, which can be found at https://github.com/IbTJensen/CompoCor. The scripts used for the simulations and data analysis can be found at https://github.com/IbTJensen/Microbiome-Cross-correlations/.

## Results

In this section, we compare the different estimation methods on simulated datasets with the correlation matrices constructed using the cluster and the loadings methods.

### Case B

Fig 1 shows the performance of the different correlation estimation methods, with correlation matrices generated by both the cluster and the loadings method. All MAEs are computed as

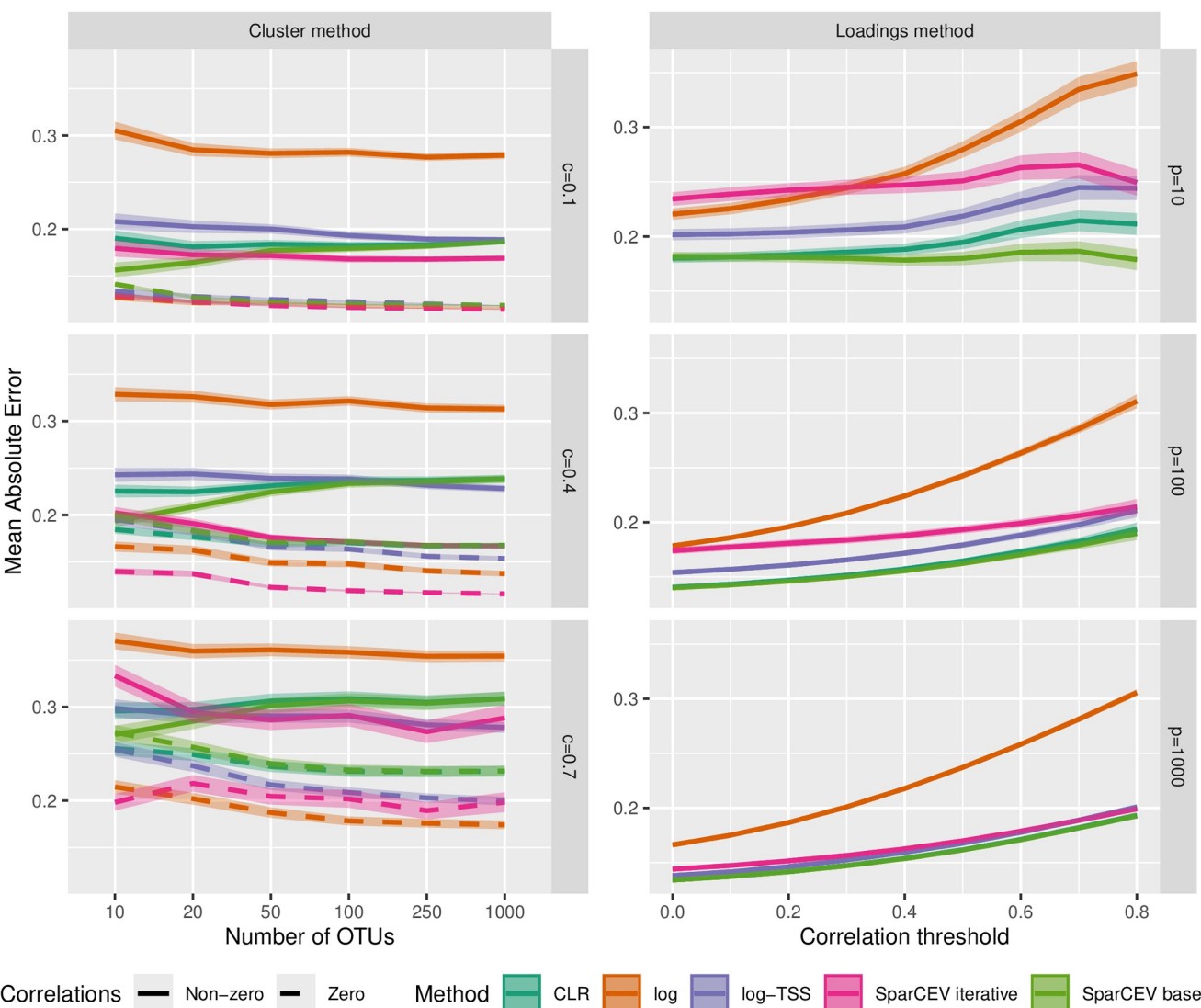

**Fig 1. Results on simulated data in case B.** MAE of different cross-correlation methods for correlation matrices generated by the cluster method (left column) and the loadings method (right column). For the cluster method, different $p$ (number of OTUs) and $c$ (the proportion of OTUs in a cluster) are used. For the loadings method, threshold values $u = 0, 0.1, \ldots, 0.8$ (cf. (7)) and different $p$ are used. The lines show the mean accuracy, and the edges of the envelopes show $\pm 1.96$ standard errors (SE). The results are based on 1000 simulated datasets where each simulated dataset has 50 replicates.

means over 1000 simulated datasets, with $n = 50$ replicates. For the cluster method $\rho = 0.75$ and for the loadings method $k = 5$ (see S1 Text). With both correlation generation methods, poor results are obtained when only the log-transformation is applied, and all other methods yield better results. For the cluster method, CLR, SparCEV base, and SparCEV iterative outperform log-TSS when $c = 0.1$, SparCEV base outperforms CLR when $p$ is small, and SparCEV iterative outperform all other methods when $p > 20$. When $c = 0.4$, log-TSS performs the same or better than CLR and SparCEV for $p \geq 100$, but SparCEV iterative substantially outperforms all other methods. This suggests that the iterative procedure successfully alleviates the bias incurred from the compositional structure relative to the other methods. For $c = 0.7$, 70% of pairs are correlated, and thus the sparsity assumption is severely violated. As expected, this is a substantial obstacle to accurate estimation, especially for CLR, SparCEV base, and SparCEV

iterative. In fact, log-TSS performs similarly or better, except when $p = 10$, where SparCEV base still has a slight edge. SparCEV iterative performs similar to log-TSS across all $p \geq 20$ for $c = 0.7$. For the loadings method, SparCEV base outperforms all alternatives when $p = 10$. For $p = 100$, the difference between SparCEV base and CLR is negligible, but both outperform log-TSS. When $p = 1000$, SparCEV base and CLR perform practically identically, and they only outperform log-TSS at higher thresholds (cf. (7)) and only by a small margin. For correlations generated by the loadings method, SparCEV iterative offers no advantage over SparCEV base, and in fact performs markedly worse when $p$ is small, although the difference shrinks as $p$ increases. This happens because the loadings method tends to produce correlations that are roughly symmetric around zero. In other words, the bias that the iterative procedure is supposed to alleviate is close to zero by construction. Consequently, the iterative procedure essentially uses less data (by excluding OTUs strongly correlated with the non-compositional variable) for no advantage. This is also why the difference shrinks as $p$ increases, as excluding some variables is less impactful when data is abundant.

The general pattern observed in Fig 1 is that log yields the worst results, log-TSS is an improvement, CLR and SparCEV base outperform log-TSS (except when sparsity is severely violated), and SparCEV base outperforms CLR at low $p$. This behavior is consistent with the theory presented in S1 Text. The performance of SparCEV iterative generally depends on the magnitude of the bias incurred by violation of the sparsity assumption. Situations where $p$ is small may be encountered in practice, for example, when abundances at high taxonomic levels are considered or when synthetic communities are employed, as is sometimes done in the plant field [24]. SparCEV iterative consistently outperforms SparCEV base for the cluster method, while the reverse is the case for the loadings method. For a real dataset, we do not know which of the scenarios the correlation structure most closely resembles. However, the advantage of SparCEV base for the loadings method is quite small for $p \geq 100$ and SparCEV base and SparCEV iterative perform almost identically for the cluster method for $p = 10$ (except when sparsity is severely violated). Our practical recommendation is thus to employ SparCEV base when $p$ is small, and SparCEV iterative otherwise. We carried out similar simulations without biological zeros. The results were practically identical and can be found in S1 Fig.

**The effect of diversity.**  Friedman and Alm [13] showed that the accuracy of correlation estimates in case A depends on the diversity of the microbiome. They show that the accuracy of empirical Pearson correlation estimates decreases as $p_{\text{eff}}$ increases, whereas SparCC is unaffected by $p_{\text{eff}}$. Fig 2 shows the impact of diversity in case B with $p = 100$ and different average $p_{\text{eff}}$. The simulation settings are identical to those in Fig 1 for $p = 100$, except that we choose $(\mu_i, \sigma_i^2)$ differently (see Parameter Selection under Material and methods). Additionally, we set $\pi_i = 0$ for all $i$ to avoid zero inflation. This is because a more zero-inflated dataset will tend to have lower entropy (and thus lower $p_{\text{eff}}$) than a less zero-inflated dataset. This introduces a chaotic element to the simulation process that may muddle the patterns we seek to investigate.

In Fig 1, we show two sets of lines for the cluster method, one for correlated pairs and one for uncorrelated pairs. In Fig 2, these lines would have fallen on top of each other, so for ease of presentation, the lines for the uncorrelated pairs have been omitted. The results for uncorrelated pairs are instead shown in S2 Fig, where the overall pattern is similar to Fig 2. However, for uncorrelated pairs, both log and log-TSS perform better than CLR and SparCEV when $p_{\text{eff}}$ is high and sparsity is violated, and SparCEV iterative outperforms all other methods under all settings.

In Fig 2, we see that SparCEV iterative, SparCEV base, and CLR are only mildly affected by the effective number of OTUs for correlation matrices generated by both the cluster method

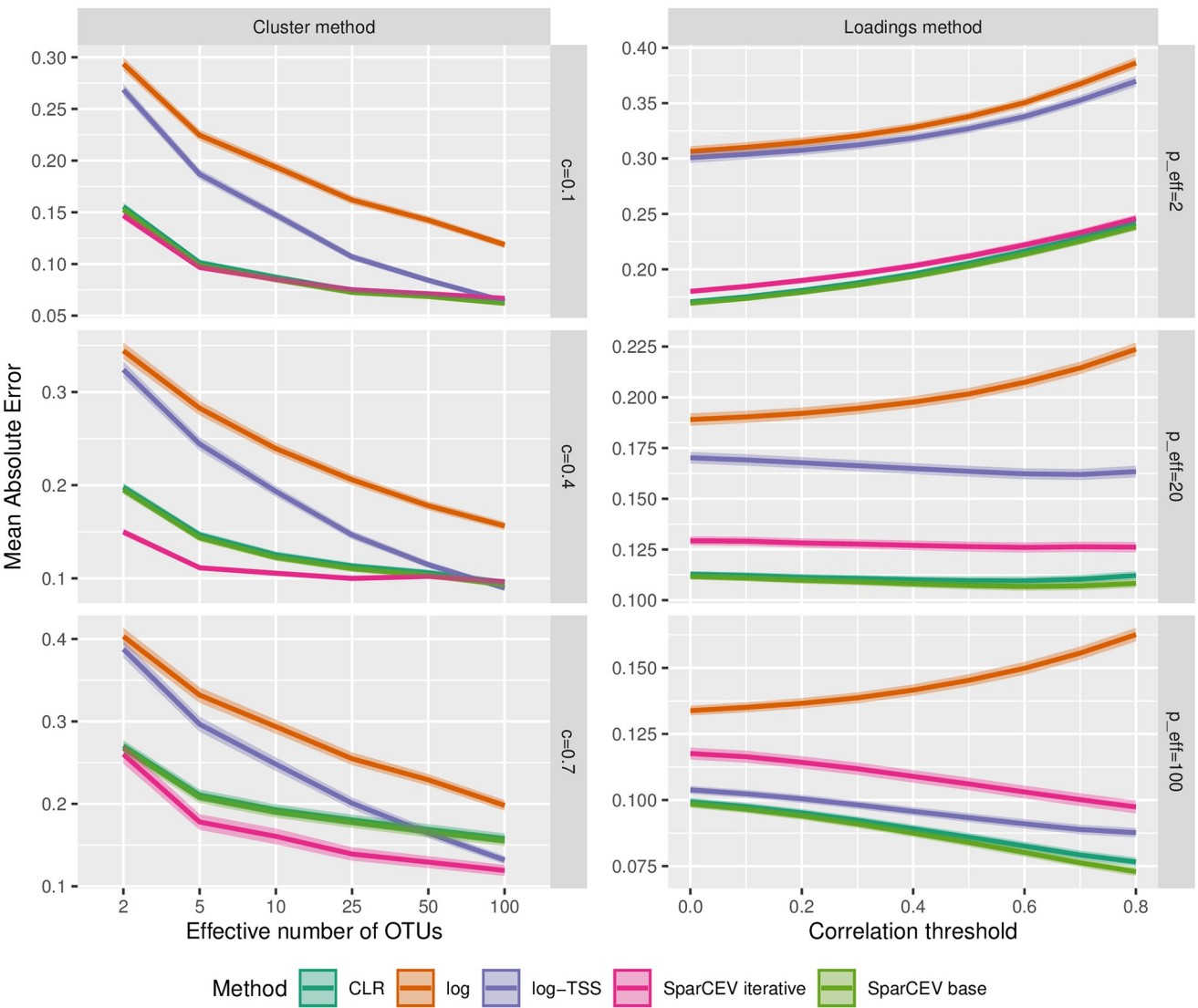

**Fig 2. Results for simulated data with differing diversity in case B.** MAE of different cross-correlation methods for correlation matrices generated by the cluster method (left column) and the loadings method (right column). For the cluster method, different $p_{eff}$ (effective number of OTUs) and $c$ (the proportion of OTUs in a cluster) are used. For the loadings method, threshold values $u = 0, 0.1, \ldots, 0.8$ (cf. (7)) and different $p_{eff}$ are used. The lines show the mean accuracy, and the edges of the envelopes show ±1.96 SE. The results are based on 1000 simulated datasets where each simulated dataset has 50 replicates.

and the loadings method, regardless of threshold (cf. (7)) or density of the correlation matrix. SparCEV base still consistently outperforms CLR, although the difference is negligible (for all $p_{eff}$, the difference is similar to the difference we saw in Fig 1 at $p = 100$). As expected, SparCEV iterative outperforms SparCEV base for the cluster method with higher levels of sparsity. The accuracy of the results obtained from log and log-TSS depend heavily on $p_{eff}$. The accuracy of log-TSS is similar to that of CLR and SparCEV only for dense correlation matrices with uniformly distributed abundances, which are unlikely to occur in nature. In general, the benefit of using CLR or SparCEV is greater for less diverse microbiota. This is consistent with established knowledge in case A [13].

**Application on atopic dermatitis data.**   In this section, we analyze the correlations found in an atopic dermatitis dataset from Byrd et al. [38, 39]. The severity of the symptoms was quantified by the widely used measure *objective SCORing of Atopic Dermatitis* (objective SCORAD) [40]. We estimate the correlations between objective SCORAD and bacterial abundances at the family level using SparCEV. The dataset contained 407 families and 27 replicates. On Fig 3D, we see the correlation coefficients estimated with SparCEV base and SparCEV iterative plotted against each other. They approximately lie on a straight line with a slope of 1 and an intercept less than zero. This is what we would expect if SparCEV base is negatively biased and SparCEV iterative successfully alleviates this bias. We obtained a correlation threshold of $m = 0.59$ using the threshold selection approach described in Materials and Methods. Additionally, we obtained bootstrap simulations by randomly permuting the SCORAD score within each replicate. These were used to calculate empirical bootstrap confidence intervals (CI) with the $BC_a$-method by Efron [41]. The families with absolute correlation with SCORAD exceeding 0.59 are shown in Fig 3A.

It is well known that colonization by *Staphylococcus aureus* can exacerbate the severity of atopic dermatitis [38]. Indeed, we find that *Staphylococcaceae* is positively correlated with the objective SCORAD (estimate: 0.76, 95%-CI: [0.60, 1.00]), see Fig 3A. Some members of the fungal family *Malasseziaceae* are believed to play a pathogenic role in atopic dermatitis [42, 43], and indeed we find this family to be positively correlated with the objective SCORAD (estimate: 0.62, 95%-CI: [0.41, 1.00]). Other studies found that the relative abundance of the genus *Propionibacterium* was depleted in patients with atopic dermatitis [44] and that the genus *Cutibacterium* may inhibit the growth of *Staphylococcus aureus* [45]. Both these genera are members of the family *Propionibacteriaceae*, but we did not find a correlation between the objective SCORAD and the abundance of this family (estimate: -0.04, 95%-CI: [-0.28, 0.64]). The strongest negative correlation detected was with the family *Hyphomicrobiaceae* (estimate: -0.63, 95%-CI: [-0.77, -0.24]), which to our knowledge does not have a previously established role in atopic dermatitis.

Fig 3B shows that the diversity is negatively correlated with the objective SCORAD score. Diversity is quantified as the effective number of families which is defined similarly as the previously considered effective number of OTUs. This is consistent with prior knowledge that the diversity of the skin microbiome is substantially reduced in atopic dermatitis patients [44, 45]. The effective number of families is only 18 (out of 407 observed families) even in the most diverse replicate (the effective number of families in the least diverse replicate is 1.2, with over 96% of the relative abundance occupied by *Staphylococcaceae*). Thus, the diversity in all replicates is low, and by Fig 2 we expect substantially more accurate correlation estimates from SparCEV or CLR compared with log-TSS.

According to Fig 3C, the estimates using log-TSS are consistently smaller than those of SparCEV. The theory in S1 Text provides a plausible explanation for this behaviour. In S1 Text, we show that

$$\mathbb{C}\mathrm{ov}\left[\log \mathrm{TSS}\left(x_i\right), b\right] = \mathbb{C}\mathrm{ov}\left[\log a_i, b\right] - \mathbb{C}\mathrm{ov}\left[\log A, b\right],$$

where $A$ denotes the microbial load. It has previously been established that S. aureus colonizes skin lesions during an atopic dermatitis flare [46]. Other studies have suggested that this is due to an increase in the absolute abundance of S. aureus rather than displacement of other microbes [47–49]. In other words, it appears that when the abundance of *Staphylococcaceae* increases during a flare, the microbial load increases along with it. As a result, the covariances (and thus also the correlations) estimated with log-TSS are negatively biased. Since SparCEV is unaffected by correlations with the microbial load, it follows that the results obtained with

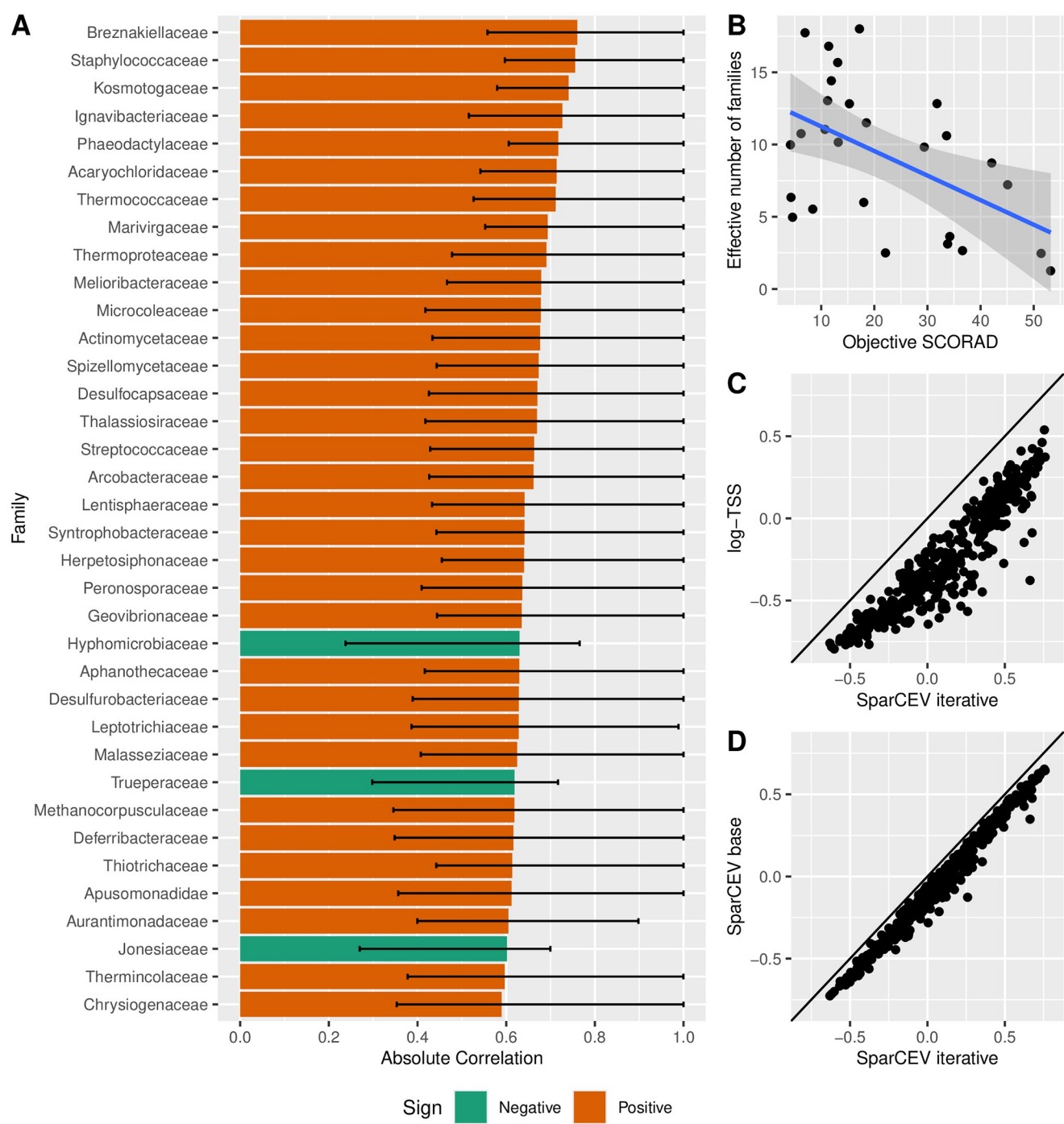

**Fig 3. Correlations between microbial abundances and the severity of atopic dermatitis.** Results from a correlation analysis on atopic dermatitis data from Byrd et al. [38]. **A**: All correlations exceeding the permutation threshold $m = 0.59$ with color according to the sign of the correlation and with error bars given by the empirical bootstrap 95%-confidence interval. **B**: Scatter plot between the effective number of families and the objective SCORAD. The blue line is derived from a smooth line fitted to the data with 95% confidence intervals derived from the standard deviation. **C**: Scatter plot between the estimated correlations using log-TSS and SparCEV. The straight line has slope 1 and intercept 0. **D**: Scatter plot between the estimated correlations using SparCEV base and SparCEV iterative. The straight line has slope 1 and intercept 0.

SparCEV are more accurate on this dataset. Our correlation estimate of the family *Malasseziaceae* is consistent with this conclusion. As noted earlier we expect a positive correlation based on previously established associations, which is what we see from SparCEV (estimate: 0.62, 95%-CI: [0.35, 1.00]), but log-TSS returns a slightly negative correlation (estimate: -0.15, 95%-CI: [-0.50, 0.25]). After applying a *t*-test to the log-TSS estimated correlations and correcting for multiple testing with Benjamini-Hochberg, 140 statistically significant correlations (at significance level 0.05) are found. Of these, 138 are negatively correlated families and with the considerations above in mind, many of these may be false positives. Comparing with SparCEV, only five families were detected by both methods (absolute correlation above *m* in SparCEV, and $p < 0.05$ after correction for multiple testing for log-TSS). These include the two families with a positive correlation coefficient by log-TSS, *Staphylococcaceae* and *Kosmotogaceae*. The other three families were *Hyphomicrobiaceae*, *Jonesiaceae*, and *Trueperaceae*, which were the only three families with a negative correlation above *m* for SparCEV.

We also applied a *t*-test to correlation coefficients estimated by CLR and compared the statistically significant correlations to those below the permutation threshold *m* of SparCEV. The *p*-values were corrected for multiple testing using Benjamini-Hochberg. In total, 76 families were statistically significant when applying CLR, while 36 were above the permutation threshold when using SparCEV. This indicates that SparCEV with permutation thresholding is more conservative. Of the 36 families above the permutation threshold, eight of them were not statistically significant when using CLR. These include families that have previously been linked to atopic dermatitis, specifically *Malasseziaceae* [42, 43] and *Streptococcaceae* [50, 51] (not to be confused with *Staphylococcaceae*, which was found by both CLR and SparCEV). Of the families found correlated with the objective SCORAD by CLR but not SparCEV (48 in total), all but one were found to be negatively correlated with the objective SCORAD. The only exception was *Casjensviridae*, which was just barely below *m* for SparCEV (Estimate: 0.57, *m* = 0.59), but just barely significant for CLR ($p = 0.047$ after correction for multiple testing). The SparCEV estimates of the remaining 47 families range between absolute values barely below *m* (e.g *Zoogloeaceae*, estimate: -0.57) and quite far from *m* (e.g *Nitrobacteraceae*, estimate -0.36). We generally find that the estimated correlation coefficients found by SparCEV are larger than those found by CLR (See S3 Fig), although the discrepancy is not as pronounced as the one seen in Fig 3C. Details on the results for all families can be found in S1 Table.

## Case C

We repeated the numerical studies from Fig 1 in case C. The left column in Fig 4 shows results with the correlation matrices obtained using the cluster method with $q = 1000$. We also carried out the simulations without biological zero, the results of which can be found in S4 Fig. For results with other values of $q = 10, 100$, see S5 Fig. On Fig 4, all tested methods perform similarly on non-correlated pairs. On correlated pairs, CLR and CLR+VST yield almost identical results and outperform log-TSS, while SparXCC base is superior when *p* or *q* is small, in agreement with the theory. On S5 Fig, we see that for $c = 0.4$, the performance lead for SparXCC is reduced and for $c = 0.7$ it is almost nonexistent. However, SparXCC iterative outperforms all other methods for $p \geq 100$ regardless of *c*. Just as in case B, a non-sparse correlation matrix is a considerable obstacle for SparXCC base, CLR, and CLR+VST. They are all outcompeted by log-TSS in this setting. However, the iterative procedure clearly alleviates the error caused by violations of sparsity, although it still performs considerably worse for $c = 0.7$ than for $c = 0.1$, 0.4.

With the loadings method, the pattern is broadly similar to the one seen in case B. When $p = q = 10$, SparXCC base outcompetes all alternatives, especially for higher thresholds (cf. (7)),

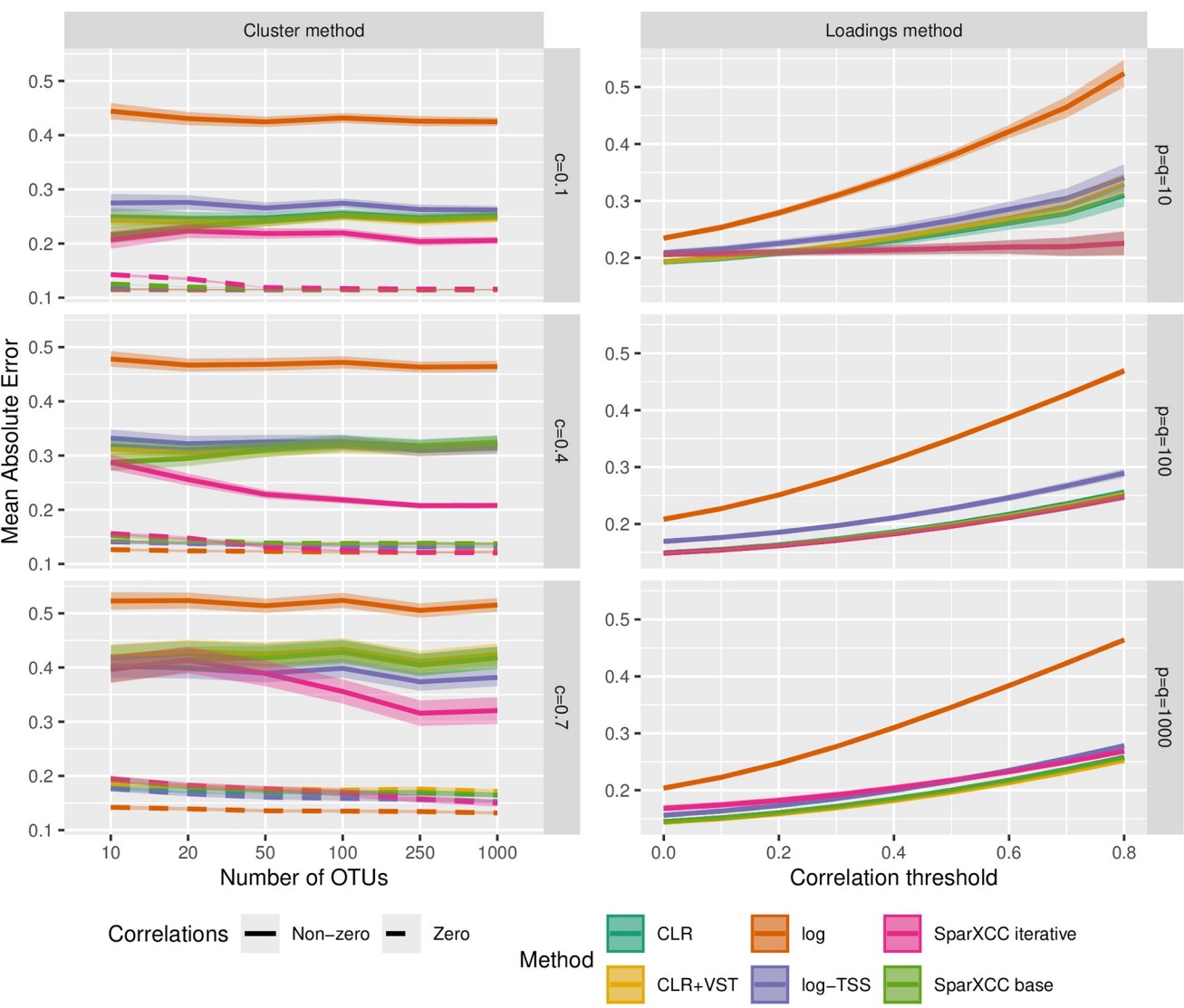

**Fig 4. Results for simulated data in case C.** MAE of different cross-correlation methods for correlation matrices generated by the cluster method (left column) and the loadings method (right column) in case C. For the cluster method, different $p$ (number of OTUs), $q$ (number of genes) and $c$ (the proportion of OTUs in a cluster) are used. For the loadings method, threshold values $u = 0, 0.1, \ldots, 0.8$ (cf. (7)) and different $p$ and $q$ are used. The lines show the mean accuracy, and the edges of the envelopes show ±1.96 standard errors (SE). The results are based on 200 simulated datasets where each simulated dataset has 50 replicates.

except SparXCC iterative, which returned identical results. This is because it typically found no OTUs or genes with average absolute correlations under the thresholds used for the iterative procedure. As a result it simply returns estimates without carrying out the iterative procedure. In other words, with this particular setting SparXCC base and SparXCC iterative are identical in many cases. For $p = q = 100$ the difference between SparXCC base, SparXCC iterative, CLR, and CLR+VST is reduced and for $p = q = 1000$, SparXCC base, CLR, and CLR+VST all perform identically and all outperform log-TSS, while SparXCC iterative performs markedly worse at low thresholds (cf. (7)). S6 Fig shows results for all tested combinations of $p$ and $q$. Situations where $p$, $q$ or both are small may arise for example when examining

correlations between 16S data (bacterial OTUs) and ITS data (fungal OTUs). Specifically, when synthetic communities are employed or when correlations at a high taxonomic level are of interest. Here SparXCC base outperforms all tested transformation-based methods when either $p$ or $q$ is sufficiently small and SparXCC iterative is identical to SparXCC base in these cases. Collectively, these results show that SparXCC outperforms the alternatives when $p$ and $q$ are small. They also show that the bias incurred by SparXCC base can be alleviated by leveraging the iterative procedure, and thus more accurate estimates can be achieved. However, in settings where this bias is small, such as for correlation matrices produced by the loadings method, SparXCC iterative may produce less accurate results than SparXCC base. On a real dataset, this may be assessed by plotting the estimates of SparXCC base and SparXCC iterative against each other. If they lie on a line with slope 1 and intercept different from 0, it indicates that SparXCC may be alleviating the bias incurred by SparXCC base. If the line has intercept 0, it indicates that SparXCC does not alleviate any bias (perhaps because none is present) and then SparXCC is preferred.

**Application on plant microbiome data.** In this section, we analyze the correlations found in the root microbiome of *Lotus japonicus* in a dataset by Thiergart et al. [20]. We have two sequencing datasets (each compositional), one from 16S ribosomal RNA and one from internal transcribed spacers (ITS). The 16S data contains bacterial OTUs, and the ITS data contains fungal OTUs. The data are from plants of multiple genotypes, the wild type (Gifu) and the mutants *ccamk*, *symrk*, *ram1*, and *nfr5*. The data contains 15–22 replicates for each genotype. We estimate the correlations using SparXCC. The replicates within each genotype originate from three different experiments. This potentially has a confounding effect on the results. For the purposes of this example, we employ the function `RemoveBatcheffect` from the R-package `limma` [34] to correct for differing means between experiments.

The next step is to assess whether to use SparXCC base or SparXCC iterative. To do this, we plotted the estimates from both methods against each other. For several of the genotypes (*ram1*, *symrk*, and *nfr5*), the results of SparXCC iterative were highly sensitive to the choice of $t_1$ and $t_2$ (see S7 Fig). Additionally, even after a suitable threshold was found, the results were on a straight line with intercept zero (see S7 Fig), indicating that the results from SparXCC base are not biased (or that SparXCC iterative fails to alleviate the biases). For these reasons, we use SparXCC base on this dataset. The results can be seen on Fig 5. More details on correlated OTUs can be found in S2 Table. A similar analysis was carried out on data collected from the rhizosphere of the plant. The results of this can be found in S8 Fig and S3 Table.

Thiergart et al. estimate cross-correlations between bacterial and fungal OTUs only on the Gifu data using Spearman correlations on TSS-transformed data (Spearman-TSS). They consider a pair correlated when the $p$-value is less than 0.001 and thus obtain 595 pairs with significant correlations. Using permutation thresholding on SparXCC, we find only 6 correlated pairs in Gifu when correcting for confounding effects. In order to make a direct comparison to the results in the original paper, we also carried out the correlation estimation without correcting for confounding effects. We then obtain 953 correlations above the permutation threshold ($m$). A substantial proportion of correlations identified by Thiergart et. al were also found by SparXCC (57%). On S9 Fig and in S4 Table, we see that in most cases, the methods find similar estimates, but in some cases they may differ considerably. In fact, in some cases, the two methods disagree on the sign of the correlation. The reason for the differences between the methods may be that SparXCC approximates Pearson correlations, which measure linear relationships, while Spearman correlations measure monotonic relationships. To examine this possibility, we also estimated the Pearson correlations of the log-TSS transformed data (Pearson-log-TSS) and found 1395 pairs with significant correlations. Interestingly, we found that Pearson-log-TSS showed a greater degree of overlap with Spearman-TSS than with SparXCC (87% vs 79%).

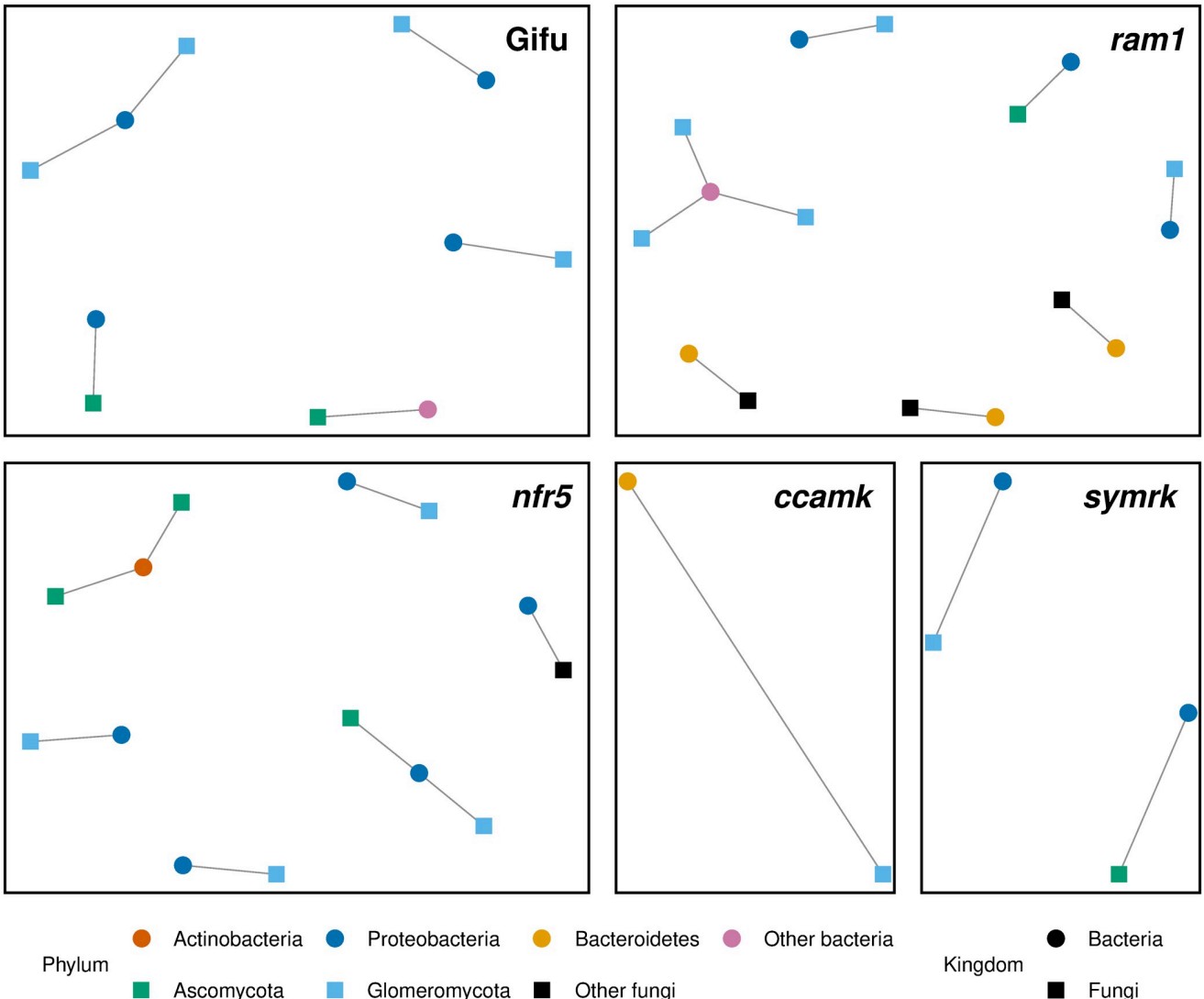

**Fig 5. Correlation network between bacterial and fungal abundances in the root of *Lotus japonicus*.** Results from applying SparXCC to 16S and ITS sequencing data from the root microbiome of *Lotus japonicus*, from Thiergart et al. [20]. Each circular vertex represents a bacterial OTU from the 16S data and a square vertex represents a fungal OTU from the ITS data. Vertices are colored based on the phylum of the OTU it represents. Two vertices are connected by an edge if their estimated correlation is above the permutation threshold. The analysis is carried out separately for the genotypes Gifu, *ram1*, *nfr5*, *ccamk*, and *symrk*. Only cross-correlations are shown.

Of the pairs where SparXCC and Spearman-TSS disagreed on the sign, 14 were above the permutation threshold but not detected as significant by the *t*-test. All of these pairs involved two specific fungal OTUs, both members of the phylum Ascomycota. Both had many reads (ranging from 95 to 2576), so these results are not an artifact of low read counts. Additionally, all of these pairs showed the same pattern when comparing SparXCC to Pearson-log-TSS; the estimated correlations had different signs, but were not detected as significant by the *t*-test. We do not know the ground truth in this data example and the associations between bacterial and fungal microbes in legumes is poorly understood. As a result, it is not possible to rely on previously established knowledge to assess which of the methods produce the most accurate results. However, our simulation study and our theoretical considerations suggest that

SparXCC produce superior results. With that in mind, this data example indicates that SparXCC may be able to capture some pattern of association that is lost with the transformation-based methods.

## Running time

We compared the running time between the different methods. In Tables 3 and 4, we see that while the running time of CLR differs substantially between cases B and C, this is not the case for SparCEV and SparXCC. This is because the most time consuming part of these methods is the variance estimation procedure adopted from SparCC. This explains why SparXCC with $p = q = 1000$ has roughly twice the running time as SparCEV with $p = 1000$ (since the variance estimation procedure has to be run twice in Case C). When one dataset is much larger than the other in case C, the running time may be completely dominated by the variance estimation of that dataset. This explains why the running time of SparXCC with $p = 1000$, $q = 10000$ is practically identical to SparCEV when $p = 10000$. Timing was carried out with the R package microbenchmark [35] on a Lenovo X1 Carbon labtop equipped with a 13th Gen Intel®Core™i7–1365U processor.

## Discussion

For the theoretical considerations in this paper, we, like Friedman and Alm [13], assume that the data follow the model in (1). According to this model, the true relative abundances $r_i$ are observed, which would only be the case with infinite sequencing depth. We nevertheless assess the different correlation estimation methods using data simulated under a more realistic setting where the $x_i$s are noisy observations of the $r_i$s. Specifically, SparseDOSSA2 assumes that the $x_i$s are multinomial, given the library size $N$ and the $r_i$s. Friedman and Alm [13] suggests mitigating the impact of the technical variance of $x_i$ given $(r_i, N)$ by using a Monte Carlo sampling procedure based on a uniform Dirichlet prior. However, in our simulation setup, we find that this reduces accuracy compared to using a pseudo-count (See S10 Fig). It is a topic of further research to investigate the nature of the technical variance (e.g. if it is truly generated by a multinomial model, as postulated by SparDossa2) and how to account for it in the cross-correlation estimations.

We did not consider testing null hypotheses of zero correlation. Due to various sources of bias, including the aforementioned technical variance, it is difficult to base hypothesis testing on theoretical results. Friedman and Alm [13] use a bootstrapping procedure when applying SparCC in case A. This is computationally demanding, however, not least since corrections for multiple testing are needed when carrying out hypothesis testing for a large number of correlations. Furthermore, in cases B and C, it is challenging to construct a bootstrap simulation scheme that respects the null hypothesis for a particular correlation while maintaining the remaining correlation structure. Due to these difficulties, we believe that it may be more appropriate to rely on the correlation estimates themselves, as we have done with the permutation threshold selection.

**Table 3. Average running time for cross-correlation estimation methods for case B in seconds.**

| p | CLR | SparCEV (base) |
|---|---|---|
| 100 | 0.0030 | 0.0047 |
| 1000 | 0.0079 | 0.210 |
| 10000 | 0.0462 | 163 |

**Table 4. Average running time for cross-correlation estimation methods for case C in seconds.**

| p | q | CLR | CLR+VST | SparXCC base |
|---|---|-----|---------|--------------|
| 1000 | 100 | 0.0144 | 0.353 | 0.190 |
| 1000 | 1000 | 0.0602 | 0.495 | 0.471 |
| 1000 | 10000 | 0.548 | 1.89 | 164 |

A fundamental assumption in this paper is that the interactions between microbes and other variables can be adequately described by a correlation matrix. To our knowledge, no alternatives have been unambiguously shown to universally better describe interactions between compositional datasets such as microbiome and RNA-seq data. Which metric is more sensible may depend on the underlying biology of the specific data under study. We compared the performance of SPIEC-EASI and correlation-based approaches in case C. SPIEC-EASI uses a penalized regression scheme to estimate the precision matrix, $\Psi^{-1}$, and does not aim to estimate the correlation matrix, $\Psi$, directly. Instead the primary aim is to discriminate between pairs that are conditionally independent and pairs that are not. With a correlation matrix constructed using the cluster method, a pair is uncorrelated if and only if it is conditionally independent (i.e $\Psi_{ij} = 0 \Leftrightarrow \Psi_{ij}^{-1} = 0$). Thus we can directly compare the power and false discovery rate (FDR) of SPIEC-EASI with those found using pair-wise correlations. To do this, we subjected the correlation estimates of CLR to a $t$-test and the estimates of SparXCC to the permutation thresholding scheme described in the material and methods section as well as using SPIEC-EASI to identify conditionally dependent pairs. The results of this comparison can be seen in S11 Fig. Compared to the thresholding method, we found that SPIEC-EASI has higher power for $n \leq 50$ but a much higher FDR when $n < 1000$. The $t$-test had similar power to SparXCC with permutation thresholding, but FDR increased as $n$ increased. However, it was able to adequately control the FDR at $n \leq 50$. Permutation thresholding saw relatively high FDR at $n = 20$, but was otherwise able to better control the FDR than the other methods. See S12 Fig for similar simulations in case B, comparing a $t$-test and permutation thresholding.

The interactions present in a real biological system are likely to be more complicated than a correlation matrix generated by the cluster method can account for. In such cases, the methods may diverge (in general, $\Psi_{ij}^{-1} = 0$ need not imply that $\Psi_{ij} = 0$ or vice versa), and it may not be clear which is more appropriate.

## Conclusion

When estimating correlations between compositional variables and non-compositional variables (case B), the results in Figs 1 and 2, and S1 Fig suggest that SparCEV iterative should be the method of choice, except when $p$ is low, in which case SparCEV base may be preferred. When estimating cross-correlations between two compositional datasets (case C), the results in Fig 4 and S4–S6 Figs suggest that the method of choice should be SparXCC base for datasets where the average cross-correlations are close to zero, and SparXCC iterative when this is not the case. In practice, this can be assessed by plotting estimates from SparXCC base and SparXCC iterative against each other. If they lie on a straight line with slope 1 and intercept different from 0, then SparXCC iterative is most likely preferable, whereas SparXCC base is preferable otherwise.

## Supporting information

**S1 Fig. Case B without biological zeros.** Accuracy of the different cross-correlation methods in case B, in the absence of biological zero by enforcing $\pi_j = 0$ for $j = 1, \ldots, p$. Otherwise, the

same simulation settings as Fig 1 are used.
(PDF)

**S2 Fig. Case B diversity and zero correlations.** Accuracy of the different cross-correlation methods in case B on uncorrelated pairs at different levels of diversity.
(PDF)

**S3 Fig. Atopic dermatitis dataset, SparCEV vs CLR.** Correlation coefficients estimated by SparCEV and CLR plotted against each other. The straight line has slope 1 and intercept 0.
(PDF)

**S4 Fig. Case C without biological zeros.** Accuracy of the different cross-correlation methods in case C, in the absence of biological zero by enforcing $\pi_j = 0$ for $j = 1, \ldots, p + q$. Otherwise, the same simulation settings as Fig 4 are used.
(PDF)

**S5 Fig. Cluster method in case C for small $q$.** Accuracy of the different cross-correlation methods on correlation matrices generated by the cluster method in case C for $q = 10, 100$. Otherwise, the same simulation settings as Fig 4 are used.
(PDF)

**S6 Fig. Loadings method in case C for all combinations of $p$ and $q$.** Accuracy of the different cross-correlation methods on correlation matrices generated by the loadings method in case C for all combinations of $p = 10, 100, 1000$ and $q = 10, 100, 1000$. Otherwise, the same simulation settings as Fig 4 are used.
(PDF)

**S7 Fig. SparXCC iterative vs SparXCC base for different thresholds.** The correlation coefficients estimated by SparXCC base and SparXCC iterative plotted against each other for both the default choice of threshold (the $80^{th}$ percentile) and a threshold chosen after manually evaluating percentiles of the permutations.
(PDF)

**S8 Fig. Cross-correlation network constructed on rhizosphere data.** Graph with edges between nodes when the cross-correlation is above a permutation threshold, estimated by SparXCC on rhizosphere data.
(PDF)

**S9 Fig. Spearman correlations of relative abundances vs SparXCC.** The estimated correlation coefficients as estimated by Spearman correlations of relative abundances plotted against correlations approximated by SparXCC. For Spearman, a pair is considered correlated when a $t$-test returns a $p$-value less than 0.001. For SparXCC, a pair is considered correlated when it is above the permutation threshold.
(PDF)

**S10 Fig. Pseudo-count versus Dirichlet Monte Carlo sampling.** Accuracy of using a pseudo-count versus Dirichlet Monte Carlo for SparCEV.
(PDF)

**S11 Fig. Separating correlated and uncorrelated pairs in Case C.** Power and FDR of CLR with a $t$-test ($p$- values corrected for multiple testing with Benjamini-Hochberg), SparXCC with permutation thresholding, and SPIEC-EASI.
(PDF)

**S12 Fig. Separating correlated and uncorrelated pairs in Case B.** Power and FDR of CLR with a $t$-test ($p$- values corrected for multiple testing with Benjamini-Hochberg) and SparCEV with permutation thresholding.
(PDF)

**S1 Table. Correlations between families and objective SCORAD score.**
(CSV)

**S2 Table. Correlations between bacterial OTUs from 16S data and fungal OTUs from ITS data from the root of *Lotus japonicus*.** Confounding experiment effects were removed and SparXCC was applied. Only pairs whose estimated correlation coefficient exceeded the permutation threshold are included.
(CSV)

**S3 Table. Correlations between bacterial OTUs from 16S data and fungal OTUs from ITS data from the rhizosphere of *Lotus japonicus*.** Confounding experiment effects were removed and SparXCC was applied. Only pairs whose estimated correlation coefficient exceeded the permutation threshold are included.
(CSV)

**S4 Table. Correlations between bacterial OTUs from 16S data and fungal OTUs from ITS data from the root of *Lotus japonicus*.** The data was not corrected for confounding effects prior to correlation estimation. The included pairs either had an correlation coefficient estimated by SparXCC exceeding the permutation threshold, or had $p < 0.001$ from a $t$-test applied to the empirical Spearman correlation of log-TSS transformed data.
(CSV)

**S1 Text. Theoretical analysis of transformation-based correlations, derivation of compositionally aware methods, and construction of correlation matrices.**
(PDF)

## Acknowledgments

We thank Adrián Gómez Repollés for assistance with the dermatitis data. We thank Thorsten Thiergart and Ruben Garrido-Oter for assistance with the plant microbiome data. We thank B Kirtley Amos and Max Gordon for critical reading. We thank Sha Zhang for supplying the data used to construct the templates for gene expression data in the simulation studies. We thank Taylor Grace FitzGerald for copy-editing.

## Author Contributions

**Conceptualization:** Ib Thorsgaard Jensen, Rasmus Waagepetersen.

**Formal analysis:** Ib Thorsgaard Jensen.

**Funding acquisition:** Simona Radutoiu.

**Project administration:** Ib Thorsgaard Jensen.

**Resources:** Simona Radutoiu.

**Software:** Ib Thorsgaard Jensen.

**Supervision:** Luc Janss, Simona Radutoiu, Rasmus Waagepetersen.

**Validation:** Ib Thorsgaard Jensen.

**Visualization:** Ib Thorsgaard Jensen.

**Writing – original draft:** Ib Thorsgaard Jensen, Rasmus Waagepetersen.

**Writing – review & editing:** Ib Thorsgaard Jensen, Rasmus Waagepetersen.

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
