## [Decision Letter · Decision Letter 0]

2 Feb 2024

PONE-D-23-34186Compositionally aware estimation of cross-correlations for microbiome dataPLOS ONE

Dear Dr. Jensen,

Thank you for submitting your manuscript to PLOS ONE. After careful consideration, we feel that it has merit but does not fully meet PLOS ONE’s publication criteria as it currently stands. Therefore, we invite you to submit a revised version of the manuscript that addresses the points raised during the review process. Please, when resubmitting your manuscript take into account the following methodological suggestions as per the Reviewer's advice (Points 1 to 4 under Comments)

1、On Real Data Analysis of Atopic Dermatitis:

2、Evaluation of the Dynamic Threshold Selection Method

3、Analysis of the Impact of Data Scale on Computational Time:

4、Specific Clarification of Multiple Comparisons Correction Method

We look forward to receiving your revised manuscript.

Kind regards,

Enrique Hernandez-Lemus, Ph.D.

Academic Editor

PLOS ONE

[Funding: This work was supported by the Bill and Melinda Gates Foundation and from

Foreign, Commonwealth & Development Office through Engineering the Nitrogen

Symbiosis for Africa (ENSA; OPP11772165). We thank Adri´an G´omez Repoll´es for

assistance with the dermatitis data. We thank Thorsten Thiergart and Ruben

Garrido-Oter for assistance with the plant microbiome data. We thank B Kirtley Amos

and Max Gordon for critical reading. We thank Sha Zhang for supplying the data used

to construct the templates for gene expression data in the simulation studies. We thank

Taylor Grace FitzGerald for copy-editing]

 [This work was supported by the Bill and Melinda Gates Foundation and from

Foreign, Commonwealth & Development Office through Engineering the Nitrogen

Symbiosis for Africa (ENSA; OPP11772165).

The funders played no role in the content of this paper.]

Reviewers' comments:

Reviewer's Responses to Questions

**Comments to the Author**

1. Is the manuscript technically sound, and do the data support the conclusions?

Reviewer #1: Yes

2. Has the statistical analysis been performed appropriately and rigorously? 

Reviewer #1: Yes

3. Have the authors made all data underlying the findings in their manuscript fully available?

Reviewer #1: Yes

4. Is the manuscript presented in an intelligible fashion and written in standard English?

Reviewer #1: Yes

5. Review Comments to the Author

Reviewer #1: Summary

In this research, the authors explore the complex domain of microbiome studies with an emphasis on deducing correlations between microbial abundances and various other variables. Addressing a notable gap in current methodologies, which primarily focus on compositional data, this paper introduces two innovative methods: SparCEV (Sparse Correlations with External Variables) and SparXCC (Sparse Cross-Correlations between Compositional data). These methods are uniquely designed to quantify correlations between OTU abundances and phenotypic variables or other compositional datasets, expanding the analytical capabilities in microbiome research. The authors have utilized a combination of real-world data analysis and comprehensive simulation studies to validate their methods.

Comments

1、On Real Data Analysis of Atopic Dermatitis:

"In the section analyzing real-world data on atopic dermatitis, it would be highly beneficial if the authors could present the highly correlated microbial species identified by other methods. A detailed comparison, particularly focusing on overlaps and distinctions among these methodologies, would greatly enhance our understanding of the uniqueness and effectiveness of your proposed approach."

2、Evaluation of the Dynamic Threshold Selection Method:

"The dynamic threshold selection method introduced in the article seems to rely significantly on subjective judgment, such as the user-defined parameter 't'. This reliance might impact the reproducibility and objectivity of the results. I would recommend that the authors explore this method in more depth, providing a more stable criterion for dynamic threshold selection or more objective guidelines to augment the universality and reliability of the method."

3、Analysis of the Impact of Data Scale on Computational Time:

"Given that the manuscript indicates similar outcomes for SparXCC and CLR transformations in Case C with large p and q values, albeit being time-consuming, a comparative analysis of the computational time across different methods as a function of data scale would be instructive. Such analysis would aid in assessing the efficiency and applicability of these methods in practical scenarios."

4、Specific Clarification of Multiple Comparisons Correction Method:

"In the process of distinguishing between correlated and uncorrelated pairs, a t-test has been applied to CLR. I would urge the authors to clearly specify the exact correction method used for addressing multiple comparisons. For example, was a Bonferroni correction or a Benjamini-Hochberg procedure employed? Clarity in this aspect is crucial for assessing the statistical rigor of the study."

6. PLOS authors have the option to publish the peer review history of their article (what does this mean?). If published, this will include your full peer review and any attached files.

Reviewer #1: No

---

## [Author Response · Author response to Decision Letter 0]

24 Apr 2024

Response to editor:

1. The manuscript has been edited to comply with the PLOS One style guidelines. Specifically,

headings has been changed to sentence case and their font sizes has been adjusted, the main text

has been changed to double space paragraph format, and the ”Support information” section has been

moved down after the references. Additionally, addresses have been removed from author affiliations

and changes in author affiliations since the time of the original submission have been incorporated.

2. Implementations of the methods SparCEV and SparXCC are now available as an R-package at

https://github.com/IbTJensen/CompoCor. Everything else necessary to replicate the results of the

paper is available at https://github.com/IbTJensen/Microbiome-Cross-correlations.

3. See reply to Editor Point 2.

4. We have removed the funding information from the Acknowledgments section in the manuscript.

Please add the following to the funding statement: ”Ib Thorsgaard Jensen and Rasmus Waagepetersen were supported by research grant VIL57389 from

Villum Fonden.”

Response to reviewer:

1. We have now added a more thorough discussion on the differences between the results found

by SparCEV, CLR and log-TSS. Additionally, a supplementary table provides the results with all three

methods on all families (this table was also included in the previous version, but it was not mentioned

in the text. This has now been rectified.).

2. Inspired by this comment, we made a refinement to the estimation procedures. Specifically,

we implemented an iterative procedure similar to the one utilized by SparCC, which both SparCEV and

SparXCC are based on. We had initially written this off after initially seeing poor results, but after more

thorough investigation, we found that it can help alleviate the bias caused by the sparsity assumption.

This obviates the need for the parameter t. We have rerun all the simulation studies and included this

new iterative approach. In some cases we see a substantial gain in accuracy, while in others we see a

decrease in accuracy (in cases where the sparsity assumption is almost exact). We also provide practical

guidance for assessing whether or not the iterative procedure is appropriate on a given dataset.

The iterative procedure makes use of user-specified thresholds, t, t1 , and t2 to select ”weakly

correlated OTUs/genes” (no connection to the t from the previous version of the manuscript). However,

we believe these are of a different nature than the t from the previous version of the manuscript

for the following reasons: Firstly, SparCC, which is already widely used in the microbiome literature,

uses a similar procedure with a similar user-specified parameter. Secondly, we suggest a bootstrap

approach to select them in a data-driven way. Thirdly, we suggest a diagnostic plot to assess whether or

not SparCEV/SparXCC with the iterative procedure provides an improvement over SparCEV/SparXCC

without it on a given dataset. In contrast, the t from the previous version of the manuscript could

neither be selected nor evaluated for a specific dataset.

3. An analysis of the running time is now included in the manuscript. Additionally, we explored a

different approach for the mathematical derivation of SparXCC. A different way to express the covari-

ances was formulated, and it was easily shown to be equivalent to the formulation from the previous

manuscript. Using this formulation substantially speeds up SparXCC. Additionally, with the iterative

procedure SparXCC can provide substantially better results than CLR in some cases, even when p and

q are large, which we believe justifies the greater running time.

4. Throughout the manuscript, all p-values are corrected for multiple testing with Benjamini-

Hochberg (except in the plant microbiome data example, where we follow the method employed in the

original paper for the purposes of comparison). This has now been clearly indicated in every instance.

---

## [Decision Letter · Decision Letter 1]

23 May 2024

Compositionally aware estimation of cross-correlations for microbiome data

PONE-D-23-34186R1

Dear Dr. Jensen,

We’re pleased to inform you that your manuscript has been judged scientifically suitable for publication and will be formally accepted for publication once it meets all outstanding technical requirements.

Kind regards,

Enrique Hernandez-Lemus, Ph.D.

Academic Editor

PLOS ONE

Additional Editor Comments (optional):

Reviewers' comments:

Reviewer's Responses to Questions

**Comments to the Author**

1. If the authors have adequately addressed your comments raised in a previous round of review and you feel that this manuscript is now acceptable for publication, you may indicate that here to bypass the “Comments to the Author” section, enter your conflict of interest statement in the “Confidential to Editor” section, and submit your "Accept" recommendation.

Reviewer #1: All comments have been addressed

2. Is the manuscript technically sound, and do the data support the conclusions?

Reviewer #1: Yes

3. Has the statistical analysis been performed appropriately and rigorously? 

Reviewer #1: Yes

4. Have the authors made all data underlying the findings in their manuscript fully available?

Reviewer #1: Yes

5. Is the manuscript presented in an intelligible fashion and written in standard English?

Reviewer #1: Yes

6. Review Comments to the Author

Reviewer #1: The authors have adeptly incorporated my feedback. I am pleased with the revisions made to the manuscript. Consequently, I wholeheartedly endorse its publication.

7. PLOS authors have the option to publish the peer review history of their article (what does this mean?). If published, this will include your full peer review and any attached files.

Reviewer #1: No

---

## [Editor Report · Acceptance letter]

19 Jun 2024

PONE-D-23-34186R1 

PLOS ONE

Dear Dr. Jensen, 

I'm pleased to inform you that your manuscript has been deemed suitable for publication in PLOS ONE. Congratulations! Your manuscript is now being handed over to our production team.

Kind regards, 

on behalf of

Prof. Enrique Hernandez-Lemus 

Academic Editor

PLOS ONE